# Triangular Fuzzy Number-Typed Fuzzy Cooperative Games and Their Application to Rural E-Commerce Regional Cooperation and Profit Sharing

**Wen-Jian Zhao [1] and Jia-Cai Liu [2],***

1   Jinshan College, Fujian Agriculture and Forestry University, Fuzhou 350002, China;
    wenjianzhao@fafu.edu.cn
2   College of Transportation and Civil Engineering, Fujian Agriculture and Forestry University,
    Fuzhou 350002, China
*   Correspondence: liujiacai@fafu.edu.cn; Tel.: +86-150-6004-2796

**Abstract:** The primary aim of this paper is to develop one kind of easy and effective method to solve fuzzy cooperative games with coalition values expressed by triangular fuzzy numbers (TFNs). This method ensures that each player should receive a TFN-typed fuzzy pay-off from the grand coalition because each coalition value is expressed by a TFN. Using the concept of Alpha-cut sets, an arbitrary TFN's Alpha-cut set can be shown as an interval. If the 1-cut sets and 0-cut sets of the TFN-typed coalition values are known, we can easily gain some important values, such as the means, the lower limits, and the upper limits of the TFN-typed payoffs via the proposed quadratic programming models and method. Furthermore, it is also easy for us to compute the lower and upper limits of Alpha-cut sets at any confidence levels of the TFN-typed payoffs for any TFN-typed cooperative game through solving the constructed quadratic programming models. Hereby the players' TFN-typed payoffs for the TFN-typed cooperative game can be explicitly solved via the representation theorem for fuzzy sets. It is easy to prove that the proposed solutions of the fuzzy cooperative games with coalition values expressed by TFNs satisfy some useful and important properties, such as symmetry, additivity, and anonymity. Finally, the validity, applicability and advantages of the proposed method is proved and discussed through a numerical example.

**Keywords:** fuzzy game theory; cooperative game; triangular fuzzy numbers; quadratic programming; algorithm

## 1. Introduction

Cooperative game theory and methodology with fuzzy coalition values have been a research hotspot in many fields such as management, economics, and business as well as environment [1–8]. The three branches of fuzzy cooperative games are shown as follows: the cooperative games whose coalitions are fuzzy [9–12], the cooperative games whose coalition values are expressed with fuzzy numbers [1–3] and the cooperative games with both fuzzy coalitions and fuzzy coalition values [4,13]. In reality, the uncertainty and fuzziness are usually expressed by fuzzy sets and/or fuzzy numbers or intuitionistic fuzzy numbers (IFNs) [14–16]. Recently, researchers started studying fuzzy cooperative and matrix games whose payoffs contain various kinds of fuzzy information. Monroy et al. [17] proposed a new model to solve the problem of cooperative fuzzy games and applied them to many real fields. Furthermore, the dominance core and preference core were established, and some characterizations were provided. Li [2] developed an easy and effective method to solve TFN-typed matrix games. Li's method was developed on the monotonicity of values in matrix games and

the computational amount and complexity are less than those of Campos's [18] and Bector et al.'s method [19]. Aggarwal et al. [20] extended the standard ranking order of Gonzalez and Vila [21] to I-fuzzy numbers and then introduced the concept of Pareto-optimal security strategies for such I-fuzzy matrix games. Li and Nang [22] developed a methodology for solving a new kind of triangular intuitionistic fuzzy number-typed matrix games. They introduced the concept of solutions for matrix games with triangular intuitionistic fuzzy number-typed payoffs and established the auxiliary intuitionistic fuzzy programming models to help players determine optimal strategies and the value of matrix games with TIFN payoffs. Kumar and Babbar [23] discussed two effective computational techniques and method to solve a kind of generalized fully fuzzy linear system which contained some triangular fuzzy numbers (TFNs). One advantage of the methods was that they successfully canceled the non-negative restriction condition on the fuzzy coefficient matrix.

As is known to all, there exist some classical solutions for crisp cooperative games, such as the Shapley value, the stable set, and the bargaining set. Nowadays, many researches are absorbed in extending the above-mentioned crisp cooperative game solutions to the field of fuzzy cooperative game and have gained abundant achievements. However, many existing fuzzy cooperative game solutions are obtained based on the operations of fuzzy numbers, especially the subtraction operations of fuzzy numbers, which will inevitably lead to the magnification of fuzziness and uncertainty. The quadratic programming model and method proposed in this paper measure the difference between the TFNs using the square of the distance instead of their subtraction operations and can effectively avoid the information distortion.

In this paper, we study the TFN-typed cooperative games based on the concept of Alpha-cut sets. In other words, the focus of this paper is to consider the kind of fuzzy cooperative games where coalition values are expressed with arbitrary TFNs. Such fuzzy cooperative games are called cooperative games with TFN-typed coalition values. Inspired by the TFNs' Alpha-cut sets and the fuzzy sets' representation theorem, we propose a simple and effective quadratic programming method for solving TFN-typed cooperative games. This method ensures that each player should receive a TFN-typed fuzzy imputation from cooperation because we suppose each coalition value is a TFN. Hereby any cooperative game with TFN-typed coalition values has TFN-typed fuzzy payoffs, which can be explicitly gained through solving the proposed quadratic programming models. The method proposed in this paper can provide analytical formulae, according to which the players' payoffs can be shown. The advantage mentioned above is remarkable and of great significance.

The rest of this paper is organized as follows. Section 2 briefly reviews some key concepts such as TFNs, Alpha-cut sets and the fuzzy sets' representation theorem for. Section 3 introduces the solution concept of cooperative games with TFN coalition values and develops two new quadratic programming methods based on Alpha-cut sets of allocations of all players to solve TFN-typed cooperative games. In Section 4, the method proposed in this paper is illustrated by a numerical example and the advantage, applicability and superiority of the proposed method are proved and discussed. Conclusion and the prospect of the future studies are made in Section 5.

## 2. TFNs and Alpha-Cut Sets

A TFN $\widetilde{a} = (a^l, a^m, a^r)$ can be regarded as a special fuzzy number [24], and its membership function is usually given by

$$\mu_{\widetilde{a}}(x) = \begin{cases} (x - a^l)/(a^m - a^l) & \text{if } a^l \leq x < a^m \\ 1 & \text{if } x = a^m \\ (a^r - x)/(a^r - a^m) & \text{if } a^m < x \leq a^r \\ 0 & \text{else,} \end{cases} \tag{1}$$

where $a^m$ is the mean of $\widetilde{a}$, $a^l$ and $a^r$ are the lower limit and the upper limit of $\widetilde{a}$, respectively. An Alpha-cut set of a TFN $\widetilde{a} = (a^l, a^m, a^r)$ is defined as $\widetilde{a}(\alpha) = \{x | \mu_{\widetilde{a}}(x) \geq \alpha\}$, where $\alpha \in [0, 1]$.

Thus, we can easily obtain an Alpha-cut set of the TFN $\tilde{a} = (a^l, a^m, a^r)$ for any $\alpha \in [0, 1]$. In the following, we will discuss the arithmetical operations and the Alpha-cuts of the TFNs.

### 2.1. The Arithmetical Operations of the TFNs

Obviously, if $a^l = a^m = a^r$, then the TFN $\tilde{a} = (a^l, a^m, a^r)$ is reduced to a real number. Conversely, real numbers are easily rewritten as TFNs. Thus, TFNs are flexible in representing various semantics of imprecision and uncertainty such as linguistics values and ill-defined quantity [25,26].

$\tilde{a} = (a^l, a^m, a^r)$ is called a non-negative TFN if $a^l \geq 0$. In this paper, we only consider non-negative TFNs. Let $\tilde{a} = (a^l, a^m, a^r)$ and $\tilde{b} = (b^l, b^m, b^r)$ be two arbitrary non-negative TFNs. Then, their arithmetical operations can be expressed as follows:

$$\tilde{a} + \tilde{b} = (a^l + b^l, a^m + b^m, a^r + b^r) \tag{2}$$

and

$$\lambda\tilde{a} = \begin{cases} (\lambda a^l, \lambda a^m, \lambda a^r) & \text{if}\lambda \geq 0 \\ (\lambda a^r, \lambda a^m, \lambda a^l) & \text{if}\lambda < 0. \end{cases} \tag{3}$$

Equations (2) and (3) mean that the sum of arbitrary non-negative TFNs and the product of an arbitrary real number and an arbitrary non-negative TFN are still TFNs.

### 2.2. Alpha-Cut Sets and the Representation Theorem

It is obvious that the Alpha-cut set of the TFN $\tilde{a} = (a^l, a^m, a^r)$ is an interval, denoted by $\tilde{a}(\alpha) = [a^L(\alpha), a^R(\alpha)]$. Particularly, $\tilde{a}(1) = a^m = [a^m, a^m]$, which can be considered as a special interval. It is easily derived from Equation (1) that $a^L(\alpha) = \alpha a^m + (1 - \alpha)a^l$ and $a^R(\alpha) = \alpha a^m + (1 - \alpha)a^r$. In particular, $\tilde{a}(1) = \{x | \mu_{\tilde{a}}(x) \geq 1\} = [a^L(1), a^R(1)] = [a^m, a^m] = a^m$ and $\tilde{a}(0) = \{x | \mu_{\tilde{a}}(x) \geq 0\} = [a^L(0), a^R(0)] = [a^l, a^r]$. According to the interval operations [27], it follows that

$$\begin{aligned} [a^L(\alpha), a^R(\alpha)] &= \left[\alpha a^m + (1 - \alpha)a^l, \alpha a^m + (1 - \alpha)a^r\right] = \alpha[a^m, a^m] + (1 - \alpha)[a^l, a^r] \\ &= \alpha\tilde{a}(1) + (1 - \alpha)\tilde{a}(0), \end{aligned} \tag{4}$$

From Equation (4), it is easily to see any Alpha-cut set of an arbitrary TFN can be directly obtained from both its 1-cut set and 0-cut set, which can be depicted as in Figure 1.

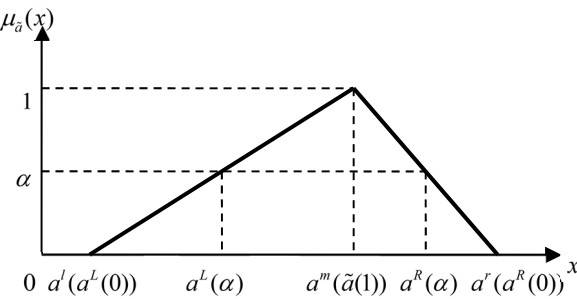

**Figure 1.** Any Alpha-cut set of a TFN $\tilde{a} = (a^l, a^m, a^r)$.

Based on the fuzzy sets' representation theorem [25] and Equation (4), an arbitrary TFN $\tilde{a} = (a^l, a^m, a^r)$ can be expressed as follows:

$$\tilde{a} = \bigcup_{\alpha \in [0,1]} \{\alpha \otimes \tilde{a}(\alpha)\} = \bigcup_{\alpha \in [0,1]} \{\alpha \otimes [\alpha\tilde{a}(1) + (1 - \alpha)\tilde{a}(0)]\}, \tag{5}$$

where $\alpha \otimes \tilde{a}(\alpha)$ is donated as a fuzzy set and its membership function is given as follows:

$$\mu_{\alpha \otimes \tilde{a}(\alpha)}(x) = \begin{cases} \alpha & \text{if } x \in \tilde{a}(\alpha) \\ 0 & \text{if otherwise,} \end{cases}$$

i.e.,

$$\mu_{\alpha \otimes \tilde{a}(\alpha)}(x) = \begin{cases} \alpha & \text{if } a^L(\alpha) \leq x \leq a^R(\alpha) \\ 0 & \text{if otherwise.} \end{cases}$$

Equation (5) means that we can easily construct a TFN if we obtain both its 1-cut set and 0-cut set.

## 3. Quadratic Programming Methods Based on Alpha-Cut Sets of TFN-Typed Coalition Values

Let us consider any fuzzy cooperative game $\tilde{v}$ with TFN-typed coalition values. A fuzzy cooperative game $\tilde{v}$ with TFN-typed coalition values on the player set $N = \{1, 2, \cdots, n\}$ means that $\tilde{v}(S)$ is a TFN for any coalition $S \subseteq N$. In particular, $\tilde{v}(S) = 0$ when $S = \varnothing$, where $\varnothing$ is an empty set. $\tilde{v}(S)$ is called the TFN-typed characteristic function of the coalition $S$. Usually, $\tilde{v}(S)$ is denoted by the TFN $\tilde{v}(S) = (a_S^l, a_S^m, a_S^r)$, where $a_S^l \leq a_S^m \leq a_S^r$ and $a_S^l \geq 0$. For convenience, $\tilde{v}(\{i\})$ and $\tilde{v}(\{i, j, \cdots\})$ are simply denoted by $\tilde{v}(i)$ and $\tilde{v}(i, j, \cdots)$, respectively. In the following, we will discuss the quadratic programming model and methods for cooperative games with coalition values expressed by TFNs based on the concept of the Alpha-cut.

### 3.1. A Quadratic Programming Model with TFN-Typed Coalition Values and Its Optimal Solution

As stated earlier, for any $\alpha \in [0, 1]$, Alpha-cut sets of payoffs $\tilde{v}(S) = (a_S^l, a_S^m, a_S^r)$ are intervals, i.e., $\tilde{v}(S)(\alpha) = \left[ \alpha a_S^m + (1 - \alpha) a_S^l, \alpha a_S^m + (1 - \alpha) a_S^r \right]$. For the sake of brevity, $\alpha a_S^m + (1 - \alpha) a_S^l$ and $\alpha a_S^m + (1 - \alpha) a_S^r$ are respectively denoted by $v^L(S)(\alpha)$ and $v^R(S)(\alpha)$ for short unless otherwise stated. For a fuzzy cooperative game $\tilde{v}$ with TFN-typed coalition values, it is obviously that each player should gain a TFN-typed imputation from the grand coalition because each coalition value is a TFN. The TFN-typed imputation of the player $i \in N$ is denoted by $x_i = (x_i^l, x_i^m, x_i^r)$.

Let $\tilde{v}(S)(\alpha) = [v^L(S)(\alpha), v^R(S)(\alpha)]$ be Alpha-cut sets of TFN-typed coalition values $\tilde{v}(S)$ and $x_i(\alpha) = [x_i^L(\alpha), x_i^R(\alpha)]$ be Alpha-cut sets of the TFN-typed imputation of the player $i \in N$, where $x_i^L(\alpha) = \alpha x_i^m + (1 - \alpha) x_i^l$ and $x_i^R(\alpha) = \alpha x_i^m + (1 - \alpha) x_i^r$. Denote $x(S)(\alpha) = \sum_{i \in S} x_i(\alpha)$, which represents the sum of Alpha-cut sets of the TFN-typed imputations of all players in the coalition $S$. Using interval operations [24], we can express $x(S)(\alpha)$ as an interval $x(S)(\alpha) = [\sum_{i \in S} x_i^L(\alpha), \sum_{i \in S} x_i^R(\alpha)]$. Distances are used to measure the difference between $x(S)(\alpha)$ and $\tilde{v}(S)(\alpha)$. Thus, according to the least square method, we define the square of the distance between the intervals $x(S)(\alpha)$ and $\tilde{v}(S)(\alpha)$ for the coalition $S$ as follows:

$$D(x(S)(\alpha), v(S)(\alpha)) = \left( \sum_{i \in S} x_i^L(\alpha) - v^L(S)(\alpha) \right)^2 + \left( \sum_{i \in S} x_i^R(\alpha) - v^R(S)(\alpha) \right)^2$$

Then, the sum of the distance squares between $x(S)(\alpha)$ and $\tilde{v}(S)(\alpha)$ for all coalitions $S$ in the grand coalition $N$ can be defined as follows:

$$\begin{aligned} L(x(\alpha)) &= \sum_{S \subseteq N} D(x(S)(\alpha), v(S)(\alpha)) \\ &= \sum_{S \subseteq N} \left[ \left( \sum_{i \in S} x_i^L(\alpha) - v^L(S)(\alpha) \right)^2 + \left( \sum_{i \in S} x_i^R(\alpha) - v^R(S)(\alpha) \right)^2 \right] \end{aligned} \tag{6}$$

where $x(\alpha) = (x_1(\alpha), x_2(\alpha), \cdots, x_n(\alpha))^T$ is the vector of Alpha-cut sets of the TFN-typed imputations for all players in the grand coalition $N$. $L(x(\alpha))$ may be interpreted as a type of loss functions.

According to the concept of loss functions, it is obvious that an optimal Alpha-cut set of allocations for all players (i.e., a solution of a TFN-typed cooperative game $v$) is the solution of the following quadratic programming model:

$$\min\{L(x(\alpha)) = \sum_{S \subseteq N} [(\sum_{i \in S} x_i^L(\alpha) - v^L(S)(\alpha))^2 + (\sum_{i \in S} x_i^R(\alpha) - v^R(S)(\alpha))^2]\} \tag{7}$$

We make the partial derivatives of $L(x(\alpha))$ in regard to the variables $x_j^L(\alpha)$ and $x_j^R(\alpha)$ ($j \in S \subseteq N$) be equal to 0, respectively. Thus, we have

$$\frac{\partial L(x(\alpha))}{\partial x_j^L(\alpha)} = 2 \sum_{S \subseteq N: j \in S} (\sum_{i \in S} x_i^L(\alpha) - v^L(S)(\alpha)) = 0 \ (j = 1, 2, \cdots, n)$$

and

$$\frac{\partial L(x(\alpha))}{\partial x_j^R(\alpha)} = 2 \sum_{S \subseteq N: j \in S} (\sum_{i \in S} x_i^R(\alpha) - v^R(S)(\alpha)) = 0 \ (j = 1, 2, \cdots, n)$$

which directly imply that

$$\sum_{S \subseteq N: j \in S} \sum_{i \in S} x_i^L(\alpha) = \sum_{S \subseteq N: j \in S} v^L(S)(\alpha) \ (j = 1, 2, \cdots, n) \tag{8}$$

and

$$\sum_{S \subseteq N: j \in S} \sum_{i \in S} x_i^R(\alpha) = \sum_{S \subseteq N: j \in S} v^R(S)(\alpha) \ (j = 1, 2, \cdots, n), \tag{9}$$

respectively.

Solving the linear equations (i.e., Equations (8) and (9)), we can get the optimal solution of the cooperative game $v$ with TFN-typed coalition values. Next, the most important thing is to solve Equations (8) and (9). For the sake of brevity, we take the solution process of Equation (8) as an example and Equation (9) can be solved in a similar fashion.

To solve $x_i^L(\alpha)$ ($i = 1, 2, \cdots, n$), Equation (8) can be rewritten as follows:

$$\begin{cases} a_{11}x_1^L(\alpha) + a_{12}x_2^L(\alpha) + a_{13}x_3^L(\alpha) + \cdots + a_{1n}x_n^L(\alpha) = \sum\limits_{S \subseteq N: 1 \in S} v^L(S)(\alpha) \\ a_{21}x_1^L(\alpha) + a_{22}x_2^L(\alpha) + a_{23}x_3^L(\alpha) + \cdots + a_{2n}x_n^L(\alpha) = \sum\limits_{S \subseteq N: 2 \in S} v^L(S)(\alpha) \\ \cdots \\ a_{n1}x_1^L(\alpha) + a_{n2}x_2^L(\alpha) + a_{n3}x_3^L(\alpha) + \cdots + a_{nn}x_n^L(\alpha) = \sum\limits_{S \subseteq N: n \in S} v^L(S)(\alpha) \end{cases} \tag{10}$$

For player $i \in N$, based on the knowledge of permutation and combination, we can know the number of the coalitions $S$ including $i$ as $C_{n-1}^0 + C_{n-1}^1 \cdots + C_{n-1}^{n-2} + C_{n-1}^{n-1}$, which is equal to $2^{n-1}$. In the similar way, the number of the coalitions $S$ including both $i$ and $j$ can be written as $C_{n-2}^0 + C_{n-2}^1 \cdots + C_{n-2}^{n-3} + C_{n-2}^{n-2}$ for players $i \in N$ and $j \in N$ ($i \neq j$), which is $2^{n-2}$. Thus, the values of $a_{ij}$ ($i, j \in \{1, 2, \cdots, n\}$) are obtained from the aforementioned conclusions as follows:

$$a_{ij} = \begin{cases} 2^{n-1} & (i = j \text{ with } i, j \in \{1, 2, \cdots, n\}) \\ 2^{n-2} & (i \neq j \text{ with } i, j \in \{1, 2, \cdots, n\}). \end{cases}$$

Denote $\boldsymbol{B}^L(\alpha) = (\sum\limits_{S \subseteq N:1 \in S} v^L(S)(\alpha), \sum\limits_{S \subseteq N:2 \in S} v^L(S)(\alpha), \cdots, \sum\limits_{S \subseteq N:n \in S} v^L(S)(\alpha))^{\mathrm{T}}$, $\boldsymbol{x}^L(\alpha) = (x_1^L(\alpha), x_2^L(\alpha), \cdots, x_n^L(\alpha))^{\mathrm{T}}$, and

$$
A = (a_{ij})_{n \times n} = \begin{pmatrix} 2^{n-1} & 2^{n-2} & \cdots & 2^{n-2} \\ 2^{n-2} & 2^{n-1} & \cdots & 2^{n-2} \\ \vdots & \vdots & & \vdots \\ 2^{n-2} & 2^{n-2} & \cdots & 2^{n-1} \end{pmatrix}_{n \times n} \tag{11}
$$

Thus, Equation (10) can be rewritten in the following matrix format:

$$
A\boldsymbol{x}^L(\alpha) = \boldsymbol{B}^L(\alpha). \tag{12}
$$

We know that the matrix $A$ is reversible through using the elementary linear transformation. Through a series of calculations, we have

$$
A^{-1} = \frac{1}{2^{n-2}} \begin{pmatrix} \frac{n}{n+1} & \frac{1}{n+1} & \cdots & -\frac{1}{n+1} \\ -\frac{1}{n+1} & \frac{n}{n+1} & \cdots & -\frac{1}{n+1} \\ \vdots & \vdots & & \vdots \\ -\frac{1}{n+1} & -\frac{1}{n+1} & \cdots & \frac{n}{n+1} \end{pmatrix}_{n \times n}
$$

By matrix multiplication, we obtain the solution of Equation (12) as follows:

$$
\boldsymbol{x}^L(\alpha) = A^{-1}\boldsymbol{B}^L(\alpha) \tag{13}
$$

In the similar way, according to the aforesaid solution method, we can obtain the following solution of Equation (9):

$$
\boldsymbol{x}^R(\alpha) = A^{-1}\boldsymbol{B}^R(\alpha) \tag{14}
$$

where $\boldsymbol{x}^R(\alpha) = (x_1^R(\alpha), x_2^R(\alpha), \cdots, x_n^R(\alpha))^{\mathrm{T}}$.

Thus, we can obtain the Alpha-cut sets of the TFN-typed imputations of the players, which are expressed as $x_i(\alpha) = [x_i^L(\alpha), x_i^R(\alpha)]$ $(i = 1, 2, \cdots, n)$. By using the above Alpha-cut sets of the TFN-typed imputations of the players, we can easily obtain the values of $x_i^L(0)$, $x_i^R(0)$, $x_i^L(1)$ and $x_i^R(1)$, where $x_i^L(1) = x_i^R(1)$. According to the fuzzy sets' representation theorem [22], the TFN-typed imputation of the player $i \in N$ can be denoted by $x_i = (x_i^l, x_i^m, x_i^r) = (x_i^L(0), x_i^L(1), x_i^R(0)) = (x_i^L(0), x_i^R(1), x_i^R(0))$.

### 3.2. A Quadratic Programming Model Considering Efficiency

In the foregoing study, we have not taken any constraint conditions into account. In this case, we consider the efficiency: $x(N)(\alpha) = \tilde{v}(N)(\alpha)$ (i.e., $[\sum\limits_{i=1}^{n} x_i^L(\alpha), \sum\limits_{i=1}^{n} x_i^R(\alpha)] = [v^L(N)(\alpha), v^R(N)(\alpha)]$), then Equation (7) can be flexibly rewritten as the following type:

$$
\min\{L(\boldsymbol{x}(\alpha)) = \sum\limits_{S \subseteq N} [(\sum\limits_{i \in S} x_i^L(\alpha) - v^L(S)(\alpha))^2 + (\sum\limits_{i \in S} x_i^R(\alpha) - v^R(S)(\alpha))^2]\}
$$

$$
\text{s.t.} \begin{cases} \sum\limits_{i=1}^{n} x_i^L(\alpha) = v^L(N)(\alpha) \\ \sum\limits_{i=1}^{n} x_i^R(\alpha) = v^R(N)(\alpha). \end{cases} \tag{15}
$$

By using the Lagrange multiplier method, we can easily construct the Lagrange function as follows:

$$
\begin{aligned}
L(\boldsymbol{x}(\alpha), \lambda(\alpha), \mu(\alpha)) = &\sum_{S \subseteq N} [(\sum_{i \in S} x_i^L(\alpha) - v^L(S)(\alpha))^2 + (\sum_{i \in S} x_i^R(\alpha) - v^R(S)(\alpha))^2] \\
&+ \lambda(\alpha)(\sum_{i=1}^{n} x_i^L(\alpha) - v^L(N)(\alpha)) + \mu(\alpha)(\sum_{i=1}^{n} x_i^R(\alpha) - v^R(N)(\alpha)).
\end{aligned}
$$

Then, Alpha-cut sets of allocations of all players (i.e., a solution of the fuzzy cooperative game $v$ with TFN-typed coalition values) are the solution of the quadratic programming model as follows:

$$
\begin{aligned}
\min\{L(\boldsymbol{x}(\alpha), \lambda(\alpha), \mu(\alpha)) = &\sum_{S \subseteq N} [(\sum_{i \in S} x_i^L(\alpha) - v^L(S)(\alpha))^2 + (\sum_{i \in S} x_i^R(\alpha) - v^R(S)(\alpha))^2] \\
&+ \lambda(\alpha)(\sum_{i=1}^{n} x_i^L(\alpha) - v^L(N)(\alpha)) + \mu(\alpha)(\sum_{i=1}^{n} x_i^R(\alpha) - v^R(N)(\alpha))\}.
\end{aligned} \tag{16}
$$

We make the partial derivatives of $L(\boldsymbol{x}(\alpha), \lambda(\alpha), \mu(\alpha))$ in regard to the variables $x_j^L(\alpha)$, $x_j^R(\alpha)$ ($j \in S \subseteq N$), $\lambda(\alpha)$ and $\mu(\alpha)$ be equal to 0, respectively. Therefore, we have

$$
\begin{cases}
\frac{\partial L(\boldsymbol{x}(\alpha), \lambda(\alpha), \mu(\alpha))}{\partial x_j^L(\alpha)} = 2 \sum_{S \subseteq N: j \in S} (\sum_{i \in S} x_i^L(\alpha) - v^L(S)(\alpha)) + \lambda = 0 \; (j = 1, 2, \cdots, n) \\
\frac{\partial L(\boldsymbol{x}(\alpha), \lambda(\alpha), \mu(\alpha))}{\partial \lambda(\alpha)} = \sum_{i=1}^{n} x_i^L(\alpha) - v^L(N)(\alpha) = 0
\end{cases}
$$

and

$$
\begin{cases}
\frac{\partial L(\boldsymbol{x}(\alpha), \lambda(\alpha), \mu(\alpha))}{\partial x_j^R(\alpha)} = 2 \sum_{S \subseteq N: j \in S} (\sum_{i \in S} x_i^R(\alpha) - v^R(S)(\alpha)) + \mu = 0 \; (j = 1, 2, \cdots, n) \\
\frac{\partial L(\boldsymbol{x}(\alpha), \lambda(\alpha), \mu(\alpha))}{\partial \mu(\alpha)} = \sum_{i=1}^{n} x_i^R(\alpha) - v^R(N)(\alpha) = 0
\end{cases}
$$

which result in

$$
\begin{cases}
\sum_{S \subseteq N: j \in S} \sum_{i \in S} x_i^L(\alpha) + \frac{\lambda(\alpha)}{2} = \sum_{S \subseteq N: j \in S} v^L(S)(\alpha) \; (j = 1, 2, \cdots, n) \\
\sum_{i=1}^{n} x_i^L(\alpha) = v^L(N)(\alpha)
\end{cases} \tag{17}
$$

and

$$
\begin{cases}
\sum_{S \subseteq N: j \in S} \sum_{i \in S} x_i^R(\alpha) + \frac{\mu(\alpha)}{2} = \sum_{S \subseteq N: j \in S} v^R(S)(\alpha) \; (j = 1, 2, \cdots, n) \\
\sum_{i=1}^{n} x_i^R(\alpha) = v^R(N)(\alpha),
\end{cases} \tag{18}
$$

respectively.

Denote $\boldsymbol{e} = (1, 1, \cdots, 1)_{n \times 1}^T$ and $\boldsymbol{x}^{L*}(\alpha) = (x_1^{L*}(\alpha), x_2^{L*}(\alpha), \cdots, x_n^{L*}(\alpha))^T$. Then, Equation (17) can be rewritten as follows:

$$
\mathbf{A}\mathbf{X}^{L*}(\alpha) + \frac{\lambda(\alpha)}{2} \boldsymbol{e} = \mathbf{B}^L(\alpha) \tag{19}
$$

and

$$
\boldsymbol{e}^T \mathbf{X}^{L*}(\alpha) = v^L(N)(\alpha) \tag{20}
$$

It follows from Equation (19) that

$$
\mathbf{X}^{L*}(\alpha) = \mathbf{A}^{-1}\mathbf{B}^L(\alpha) - \frac{\lambda(\alpha)}{2}\mathbf{A}^{-1}\boldsymbol{e} = \mathbf{X}^L(\alpha) - \frac{\lambda(\alpha)}{2}\mathbf{A}^{-1}\boldsymbol{e}, \tag{21}
$$

where $\mathbf{X}^L(\alpha)$ is given by Equation (13). Then, the most important process of solving Equation (17) is to calculate the value of $\lambda(\alpha)$.

We can easily deduce from Equations (20) and (21) that

$$e^{\mathrm{T}}x^{L}(\alpha) - \frac{\lambda(\alpha)}{2}e^{\mathrm{T}}A^{-1}e = v^{L}(N)(\alpha).$$

Obviously, we have

$$e^{\mathrm{T}}x^{L}(\alpha) = \sum_{i=1}^{n} x_i^{L}(\alpha)$$

and

$$e^{\mathrm{T}}A^{-1}e = \frac{1}{2^{n-2}}\frac{n}{n+1}.$$

Hence, we have

$$\frac{\lambda(\alpha)}{2} = 2^{n-2}\frac{n+1}{n}(\sum_{i=1}^{n} x_i^{L}(\alpha) - v^{L}(N)(\alpha)). \tag{22}$$

Thus, conclusion can be easily drawn from Equations (21) and (22) that

$$\begin{aligned}
x^{L*}(\alpha) &= x^{L}(\alpha) - 2^{n-2}\tfrac{n+1}{n}(\sum_{i=1}^{n} x_i^{L}(\alpha) - v^{L}(N)(\alpha))A^{-1}e \\
&= x^{L}(\alpha) - 2^{n-2}\tfrac{n+1}{n}(\sum_{i=1}^{n} x_i^{L}(\alpha) - v^{L}(N)(\alpha))(\tfrac{1}{2^{n-2}}\tfrac{1}{n+1})e \\
&= x^{L}(\alpha) - \tfrac{1}{n}(\sum_{i=1}^{n} x_i^{L}(\alpha) - v^{L}(N)(\alpha))e.
\end{aligned}$$

Namely,

$$x^{L*}(\alpha) = x^{L}(\alpha) + \frac{1}{n}(v^{L}(N)(\alpha) - \sum_{i=1}^{n} x_i^{L}(\alpha))e. \tag{23}$$

where $x^{L*}(\alpha) = (x_1^{L*}(\alpha), x_2^{L*}(\alpha), \cdots, x_n^{L*}(\alpha))^{\mathrm{T}}$.

In a similar way, the solution of Equation (18) can be obtained as follows:

$$x^{R*}(\alpha) = x^{R}(\alpha) + \frac{1}{n}(v^{R}(N)(\alpha) - \sum_{i=1}^{n} x_i^{R}(\alpha))e. \tag{24}$$

where $x^{R*}(\alpha) = (x_1^{R*}(\alpha), x_2^{R*}(\alpha), \cdots, x_n^{R*}(\alpha))^{\mathrm{T}}$.

So far, we obtain the solution of Equation (15) as Equations (23) and (24). Thus, if we consider the efficiency, the Alpha-cut sets of allocations of all players (i.e., a solution of a TFN-typed cooperative game $v$) can be determined as $x_i{}^{*}(\alpha) = [x_i^{L*}(\alpha), x_i^{R*}(\alpha)]$ ($i = 1, 2, \cdots, n$), with the lower limit and the upper limit given by Equations (23) and (24), respectively.

As stated in Section 2.2, by using the fuzzy sets' representation theorem [22], the TFN-typed imputation of the player $i \in N$ can be expressed as $x_i{}^{*} = (x_i^{l*}, x_i^{m*}, x_i^{r*}) = (x_i^{L*}(0), x_i^{L*}(1), x_i^{R*}(0)) = (x_i^{L*}(0), x_i^{R*}(1), x_i^{R*}(0))$.

In what follows, we will briefly describe one simple and effective algorithm for the optimal solution of Equation (15). In general, we consider an arbitrary fuzzy cooperative game $\tilde{v}$ with TFN-typed coalition values where $\tilde{v}(i) = (0, 0, 0)$ for all $i \in N$.

Denote an ordered array $(x^{Lk}, M^{Lk})$ ($k = 1, 2, 3, \ldots$), where $x^{Lk}$ is a vector of players' lower bounds of the Alpha-cut sets of the TFN-typed imputation and $M^{Lk} \in N$. The schema of the algorithm processes can be shown as follows:

(1)   $x^{L1} = x^{L*}(\alpha)$
(2)   $M^{L1} = \{j \in N / x_i^{L*}(\alpha) < 0\}$; $M^{L0} = \varnothing$

(3)   $x_i^{L(k+1)} = \begin{cases} x_i^{Lk} + \frac{x^{Lk}(M^{Lk})}{n-m_k^L} & (\forall j \notin M^{Lk}) \\ 0 & (\forall j \in M^{Lk}) \end{cases}$ , where $m_k^L$ denotes the size of $M^{Lk}$ and $M^{L(k+1)} =$

$M^{Lk} \cup \{j \in N / x_i^{L(k+1)} < 0\}$

(4)   The algorithm processes stop once $M^{Lk} = M^{L(k-1)}$.

In the similar way to the lower bounds of the Alpha-cut sets of the TFN-typed imputation of the player $i \in N$ (i.e., Algorithm 1), we can easily obtain the corresponding upper bounds which is omitted here.

## 4. A Numerical Example and Computational Result Analysis

Section 3 carefully discusses the solution concept and solution method of the TFN-typed cooperative games. It is easy to see that determination of each player's coalition value is a key to applying the proposed method for solving decision problems in real fields, such as e-commerce, logistics, management, investment, and environment. In this section, we employ a numerical example to prove and discuss the advantage, applicability, and superiority of the proposed method. The example will use the proposed TFN-typed cooperative game theory and method based on the Alpha-cut sets to determine optimal allocation strategies.

Assume that there are three rural e-commerce firms (i.e., players) 1, 2 and 3, each of whom can conduct business independently. Denote the grand coalition by $N = \{1, 2, 3\}$. To improve economic performance, they plan to cooperate. It is difficult for them to forecast precisely their profits owing to the complexity of market and the uncertainty of information. However, they can estimate the approximate ranges and the membership degrees of their profits, which can be conveniently expressed by TFNs. In this case, if they conduct business by themselves, they have the same profits which are expressed with the TFNs $\widetilde{v}(1) = \widetilde{v}(2) = \widetilde{v}(3) = (4, 8, 10)$. Similarly, if any two firms cooperate, then their profits are expressed as $\widetilde{v}(1,2) = (25, 30, 40)$, $\widetilde{v}(1,3) = (22, 28, 45)$ and $\widetilde{v}(2,3) = (25, 35, 50)$, respectively. If all the three firms (i.e., the grand coalition $N$) cooperate, then the profit is expressed as $\widetilde{v}(1,2,3) = (80, 100, 120)$.

### 4.1. Computational Results Obtained by the Proposed Method

By simple calculation, $\sum\limits_{S \subseteq N: i \in S} v^L(S)(\alpha)$ $(i = 1, 2, 3)$ can be obtained as follows:

$\sum\limits_{S \subseteq N: 1 \in S} v^L(S)(\alpha) = 8\alpha + 4(1-\alpha) + 30\alpha + 25(1-\alpha) + 28\alpha + 22(1-\alpha) + 100\alpha + 80(1-\alpha) = 35\alpha + 131$, $\sum\limits_{S \subseteq N: 2 \in S} v^L(S)(\alpha) = 8\alpha + 4(1-\alpha) + 30\alpha + 25(1-\alpha) + 35\alpha + 25(1-\alpha) + 100\alpha + 80(1-\alpha) = 39\alpha + 134$, and $\sum\limits_{S \subseteq N: 3 \in S} v^L(S)(\alpha) = 8\alpha + 4(1-\alpha) + 28\alpha + 22(1-\alpha) + 35\alpha + 25(1-\alpha) + 100\alpha + 80(1-\alpha) = 40\alpha + 131$, respectively. Thus,

$$\boldsymbol{B}^L(\alpha) = \begin{pmatrix} \sum\limits_{S \subseteq N: 1 \in S} v^L(S)(\alpha) \\ \sum\limits_{S \subseteq N: 2 \in S} v^L(S)(\alpha) \\ \sum\limits_{S \subseteq N: 3 \in S} v^L(S)(\alpha) \end{pmatrix} = \begin{pmatrix} 35\alpha + 131 \\ 39\alpha + 134 \\ 40\alpha + 131 \end{pmatrix}.$$

It is easily derived from Equation (13) that

$$\boldsymbol{x}^L(\alpha) = A^{-1}\boldsymbol{B}^L(\alpha) = \begin{pmatrix} \frac{3}{8} & -\frac{1}{8} & -\frac{1}{8} \\ -\frac{1}{8} & \frac{3}{8} & -\frac{1}{8} \\ -\frac{1}{8} & -\frac{1}{8} & \frac{3}{8} \end{pmatrix} \begin{pmatrix} 35\alpha + 131 \\ 39\alpha + 134 \\ 40\alpha + 131 \end{pmatrix} = \begin{pmatrix} \frac{13}{4}\alpha + 16 \\ \frac{21}{4}\alpha + \frac{35}{2} \\ \frac{23}{4}\alpha + 16 \end{pmatrix}.$$

In a similar way, using Equation (14), we can obtain $x^R(\alpha)$ as follows:

$$x^R(\alpha) = A^{-1}B^R(\alpha) = \begin{pmatrix} \frac{3}{8} & -\frac{1}{8} & -\frac{1}{8} \\ -\frac{1}{8} & \frac{3}{8} & -\frac{1}{8} \\ -\frac{1}{8} & -\frac{1}{8} & \frac{3}{8} \end{pmatrix} \begin{pmatrix} -49\alpha + 215 \\ -47\alpha + 220 \\ -54\alpha + 225 \end{pmatrix} = \begin{pmatrix} -\frac{23}{4}\alpha + 25 \\ -\frac{19}{4}\alpha + \frac{55}{2} \\ -\frac{33}{4}\alpha + 30 \end{pmatrix}$$

Thus, we have

$$x_2(\alpha) = [x_2^L(\alpha), x_2^R(\alpha)] = [\frac{21}{4}\alpha + \frac{35}{2}, -\frac{19}{4}\alpha + \frac{55}{2}]$$

$$x_1(\alpha) = [x_1^L(\alpha), x_1^R(\alpha)] = [\frac{13}{4}\alpha + 16, -\frac{23}{4}\alpha + 25],$$

and

$$x_3(\alpha) = [x_3^L(\alpha), x_3^R(\alpha)] = [\frac{23}{4}\alpha + 16, -\frac{33}{4}\alpha + 30].$$

So far, we have obtained the Alpha-cut sets of the TFN-typed imputations of the players. Particularly, when $\alpha = 0$ and $\alpha = 1$, we have:

$$x_1^L(0) = 16, \ x_1^R(0) = 25, \ x_1^L(1) = \frac{77}{4} \text{ and } x_1^R(1) = \frac{77}{4}.$$

According to the fuzzy sets' representation [22], the TFN-typed imputation of player 1 can be obtained as $x_1 = (x_1^l, x_1^m, x_1^r) = (x_1^L(0), x_1^L(1), x_1^R(0)) = (16, \frac{77}{4}, 25)$.

Similarly, we can obtain the TFN-typed imputations of player 2 and player 3 as $x_2 = (\frac{35}{2}, \frac{91}{4}, \frac{55}{2})$ and $x_3 = (16, \frac{87}{4}, 30)$, depicted as in Figure 2.

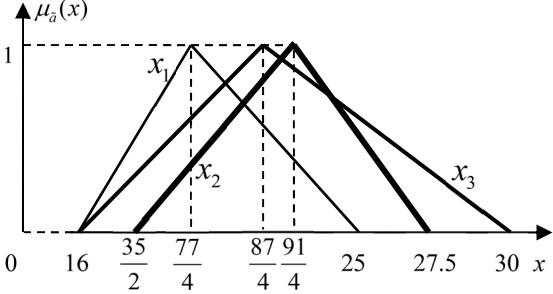

**Figure 2.** The optimal TFN-typed imputations of players 1, 2 and 3.

As stated earlier, the above results do not consider the efficiency condition. In the following, we obtain the optimal allocation strategy based on Equations (23) and (24). In this case, $v^L(N)(\alpha) - \sum_{i=1}^{3} x_i^L(\alpha)$ and $v^R(N)(\alpha) - \sum_{i=1}^{3} x_i^R(\alpha)$ can be calculated as follows:

$$\begin{aligned} v^L(N)(\alpha) - \sum_{i=1}^{3} x_i^L(\alpha) &= 100\alpha + 80(1-\alpha) - [(\frac{13}{4}\alpha + 16) + (\frac{21}{4}\alpha + \frac{35}{2}) + (\frac{23}{4}\alpha + 16)] \\ &= \frac{23}{4}\alpha + \frac{61}{2} \end{aligned}$$

and

$$\begin{aligned} v^R(N)(\alpha) - \sum_{i=1}^{3} x_i^R(\alpha) &= 100\alpha + 120(1-\alpha) - [(-\frac{23}{4}\alpha + 25) + (-\frac{19}{4}\alpha + \frac{55}{2}) + (-\frac{33}{4}\alpha + 30)] \\ &= -\frac{5}{4}\alpha + \frac{75}{2}, \end{aligned}$$

respectively. Thus, it easily follows from Equations (22) and (23) that

$$\boldsymbol{x}^{L*}(\alpha) = \boldsymbol{x}^L(\alpha) + \tfrac{1}{3}(v^L(N)(\alpha) - \sum_{i=1}^{3} x_i^L(\alpha))e = \begin{pmatrix} \frac{13}{4}\alpha + 16 \\ \frac{21}{4}\alpha + \frac{35}{2} \\ \frac{23}{4}\alpha + 16 \end{pmatrix} + \tfrac{1}{3}(\tfrac{23}{4}\alpha + \tfrac{61}{2})\begin{pmatrix} 1 \\ 1 \\ 1 \end{pmatrix} = \begin{pmatrix} \frac{31}{6}\alpha + \frac{157}{6} \\ \frac{43}{6}\alpha + \frac{83}{3} \\ \frac{23}{3}\alpha + \frac{157}{6} \end{pmatrix}$$

and

$$\boldsymbol{x}^{R*}(\alpha) = \boldsymbol{x}^R(\alpha) + \tfrac{1}{3}(v^R(N)(\alpha) - \sum_{i=1}^{3} x_i^R(\alpha))e = \begin{pmatrix} -\frac{23}{4}\alpha + 25 \\ -\frac{19}{4}\alpha + \frac{55}{2} \\ -\frac{33}{4}\alpha + 30 \end{pmatrix} + \tfrac{1}{3}(-\tfrac{5}{4}\alpha + \tfrac{75}{2})\begin{pmatrix} 1 \\ 1 \\ 1 \end{pmatrix} = \begin{pmatrix} -\frac{37}{6}\alpha + \frac{75}{2} \\ -\frac{31}{6}\alpha + 40 \\ -\frac{26}{3}\alpha + \frac{85}{2} \end{pmatrix},$$

respectively.

By using the fuzzy sets' representation theorem [25], the TFN-typed imputations of the players $i \in N$ can be expressed as follows:

$$x_1^* = (x_1^{l*}, x_1^{m*}, x_1^{r*}) = (x_1^{L*}(0), x_1^{L*}(1), x_1^{R*}(0)) = (\frac{157}{6}, \frac{94}{3}, \frac{75}{2}),$$

$$x_2^* = (x_2^{l*}, x_2^{m*}, x_2^{r*}) = (x_2^{L*}(0), x_2^{L*}(1), x_2^{R*}(0)) = (\frac{83}{3}, \frac{209}{6}, 40)$$

and

$$x_3^* = (x_3^{l*}, x_3^{m*}, x_3^{r*}) = (x_3^{L*}(0), x_3^{L*}(1), x_3^{R*}(0)) = (\frac{157}{6}, \frac{203}{6}, \frac{85}{2}),$$

respectively, depicted as in Figure 3.

It is easily seen that the above TFN-typed allocations of all players are more reasonable than those without considering the efficiency. The sums of the means, the lower and upper limits of the above TFN-typed allocations of all players are

$$\frac{157}{6} + \frac{83}{3} + \frac{157}{6} = 80,$$

$$\frac{94}{3} + \frac{209}{6} + \frac{203}{6} = 100$$

and

$$\frac{75}{2} + 40 + \frac{85}{2} = 120,$$

respectively, which are equal to the mean, the lower and upper limits of the TFN-typed value of the grand coalition $N$. Moreover, the quadratic programming model and method considering the efficiency can guarantee that the payoffs of the grand coalition $N$ are distributed thoroughly at any membership degree.

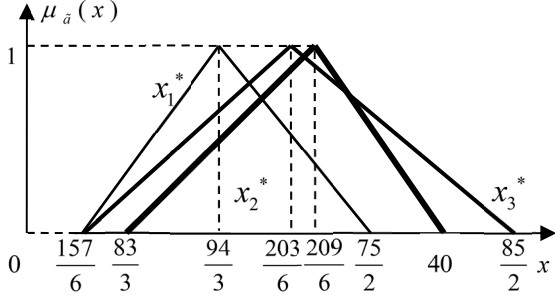

**Figure 3.** The TFN-typed imputations of players 1, 2 and 3 considering efficiency.

*4.2. Discussion and the Superiority of the Proposed Method*

According to the computational results obtained by the quadratic programming model considering efficiency, all the three players can gain the satisfactory profit allocation values. Taking play 1 for example, $x_1{}^* = \left(\frac{157}{6}, \frac{94}{3}, \frac{75}{2}\right)$ means the mean and the most possible pay-off for play 1 is $\frac{94}{3}$, the lower limit of the possible pay-off for play 1 is $\frac{157}{6}$, and the upper limit of the possible pay-off for play 1 is $\frac{75}{2}$. If play 1 delivery parcels independently instead of cooperation, the mean profit will be 8, the lower limit of the possible pay-off will be 4, and the upper limit of the possible pay-off will be 10. Obviously, player 1 will yield more benefit via cooperating with players 2 and 3. $x_2{}^* = \left(\frac{83}{3}, \frac{209}{6}, 40\right)$ and $x_3{}^* = \left(\frac{157}{6}, \frac{203}{6}, \frac{85}{2}\right)$ have the similar definition.

Comparing and analyzing the aforementioned quadratic programming methods and computational results, the following advantages and superiority of the proposed method can be easily seen:

(1)  Rationality. In many real management situations, the prospective returns of cooperation are inevitably imprecise or not totally reliable owing to the limitations of human expertise, experience, and knowledge. The TFN-typed value can appropriately express the uncertainty and fuzziness. The models and methods proposed in this paper can effectively solve the TFN-typed cooperative games.

(2)  Superiority. In this paper, we develop an easy and effective way to solve TFN-typed cooperative games based on the quadratic programming method and the square distance, which can bring down the uncertainty magnification and information distortion to a great extent.

(3)  Computational complexity. The proposed model and method in this paper are simpler and more convenient than other methods in term of the computational complexity. Players' TFN-typed payoffs can be obtained simultaneously through the proposed method in this paper. However, other methods can only be used to solve the imputations of players one by one.

## 5. Conclusions

We develop quadratic programming models and methods to solve TFN-typed cooperative games based on Alpha-cut sets and the fuzzy sets' representation theorem. Using the proposed method in this paper, any TFN-typed cooperative game has the sole optimal solution, which can be explicitly obtained by solving the quadratic programming models (i.e., Equation (7) or Equation (15)). In some respects, such as the efficiency, calculation complexity and the perform of algorithm, the proposed method in this paper have an obvious advantage.

In this paper, we use TFNs to show the uncertainty and imprecision in the real world and study how to solve fuzzy cooperative games with TFN-typed coalition values. However, trapezoidal fuzzy numbers [28–31] and intuitionistic fuzzy numbers can also be used to characterize fuzziness and uncertainty at some point. Thus, we will develop some effective models and methods to solve cooperative games with coalition values represented by trapezoidal fuzzy numbers and/or intuitionistic fuzzy numbers in the near future.

**Author Contributions:** W.-J.Z. conceived, designed, and wrote the manuscript; J.-C.L. collected the data, and revised the manuscript. Both the authors have read and approved the final manuscript.

**Funding:** This research was funded by the Social Science Planning Program of Fujian Province of China (No. FJ2018B014) and the National Natural Science Foundation of China (No. 71572040). The APC was funded by Fujian Agriculture and Forestry University.

**Conflicts of Interest:** The authors declare no conflict of interest.

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
