# Peer review of "Triangular Fuzzy Number-Typed Fuzzy Cooperative Games and Their Application to Rural E-Commerce Regional Cooperation and Profit Sharing"

_symmetry, doi:10.3390/sym10120699_

Round 1
Reviewer 1 Report
The authors develop quadratic programming models and methods for solving fuzzy cooperative games using triangular fuzzy numbers. I have checked the paper and I found it correct. The only thing that I have not checked is the numerical example. I would suggest in a future work to try use trapezoid fuzzy number instead of triangular ones. I believe that would be even more important.
I believe that this work would be published ion the present form.
Reviewer 2 Report
In this paper is proposed a novel method for solving fuzzy cooperative games with coalition values given by triangular fuzzy numbers.
In the introduction section authors must highlight the aims of their research, the limits of the recent works in literature and the advantages in terms of performance of the proposed method.
In section 3 authors must add a schema of the processes in their method and present their algorithm in a pseudocode form.
A deeper discussion of the computational results in paragraph 4.2 is necessary.
Author Response
Dear reviewer 2,
Thanks very much for all of your constructive and excellent comments. All of them have been taken into consideration and we have made careful revisions according to your requirements.
All revisions have been clearly highlighted using the “Track Changes” function in Microsoft Word in the revised manuscript. We have explained point-by-point the details of the revisions in the manuscript and the responses to the your comments.

Reviewer 3 Report
1. “alfa” should be “alpha” across the text. 2. Please add introductory notes at the beginning of each section (e.g. Section 2) to discuss its contents. 3. Please explain why quadratic programming model is needed if other approaches can be taken, e.g. Shapley value in https://www.tandfonline.com/doi/full/10.1080/02331934.2014.956743 4. Please clearly define the result (variables) of the calculations (these are pay-offs for players). 5. It is unclear whether quadratic programming is used in Section 4. 6. The title of Section 4.2 is incorrect. 7. The following phrase is incorrect: “take the efficiency into account to reallocate profits more reasonable.”Author Response
Dear reviewer 3,
Thanks very much for all of your constructive and excellent comments. All of them have been taken into consideration and we have made careful revisions according to your requirements.
All revisions have been clearly highlighted using the “Track Changes” function in Microsoft Word in the revised manuscript. In the following, we have explained point-by-point the details of the revisions in the manuscript and the responses to the reviewers’ comments.

Round 2
Reviewer 2 Report
In this new version of their manuscript authors take in account all my suggestions. I consider this paper publishable in the corrente form.
Author Response
Dear reviewer,
Thanks very much for your support and comments.
Best wishes,
Jia-cai Liu

Reviewer 3 Report
English check is needed, e.g. "palyers"
Author Response
Dear reviewer,
Thanks very much for your kind support and valued comments. We have revised carefully according to your suggestion.
Best wishes,
Wen-jian Zhao, Jia-cai Liu
